# Opiate Prescriptions Vary among Common Urologic Procedures: A Claims Dataset Analysis

**DOI:** 10.3390/jcm11051329

**Published:** 2022-02-28

**Authors:** Anish B. Patel, Praveen N. Satarasinghe, Victoria Valencia, Jessica L. Wenzel, Jack C. Webb, J. Stuart Wolf, E. Charles Osterberg

**Affiliations:** Department of Surgery and Perioperative Care, Dell Seton Medical Center, The University of Texas at Austin, Austin, TX 78712, USA; praveensatarasinghe@utexas.edu (P.N.S.); victoria.valencia@austin.utexas.edu (V.V.); jessicalr@utexas.edu (J.L.W.); jack_webb@utexas.edu (J.C.W.); stuart.wolf@austin.utexas.edu (J.S.W.J.); eosterberg@ascension.org or

**Keywords:** opioid, prescription, pain, urology

## Abstract

Objectives: This study aimed to better understand differences in the total days’ supply and fills of common opiates following urologic procedures. Materials and Methods: The Truven Health MarketScan^®^ database was used to extract CPT codes from adults 18 years or older who underwent a urologic procedure with 90-day follow-up from 2012–2015 within the Austin–Round Rock, Texas metropolitan service area. A multivariate analysis and first hurdle modeling with a logistic outcome for any opiates was used to (1) assess differences in opioid prescribing patterns, (2) investigate opioid prescription outcomes, and (3) explore variability among opiate prescription patterns across seven urologic procedure categories. Results: Among the 2312 patients who met the inclusion criteria, 23.7% received an opiate, with an average total day’s supply of 6.20 (range 2.61–10.59). The proportion of patients receiving opiates varied significantly by procedure type (*p* = 0.028). Patients that had reconstructive procedures had the highest proportion of any opiates and the highest number of mean opiate prescriptions among the seven procedure categories (42% received opiates, *p* = 0.028, mean opiate prescriptions were 1.0 among all patients, *p* = 0.026). After adjustments, the multivariate analysis demonstrated that patients undergoing reconstructive procedures filled more opiate prescriptions (odds ratio (OR) = 1.86, 95% confidence interval (CI) = 1.00–3.50, *p* = 0.05) compared to other subcategories. Of those that received opiates, reconstructive patients had a shorter time to fills (mean −18.4 days, CI −8.40 to −28.50, *p* < 0.001). Conclusion: Patients undergoing reconstructive procedures are prescribed and fill more opiates compared to other common urological procedures. The standardization and implementation of postoperative pain regimens may help curtail this variability.

## 1. Introduction

There are more than 650,000 opioid prescriptions written each day in the United States, equating to nearly one prescription for every adult annually in the general population [1,2]. From 2000–2018, nearly a quarter of a million people have died in the United States from overdoses related to prescription opioids, with a five-fold increase in overdose deaths during this time period [3,4,5]. Despite the growth in opioid prescription volume, the amount of pain that Americans report has not decreased significantly [5]. The increase in opiate prescription volume is partially attributed to the discrepancy in opioid prescription patterns among practitioners and is influenced by factors such as physician personal preference, variability between healthcare systems, state legislation, and regional opiate availability [6,7].

The opioid crisis is a significant issue within surgery, with opioids accounting for 40% of surgical prescriptions [8]. Many studies have reported that surgical opioid prescription patterns exceed state guidelines and vary between facilities and regions [6,9,10]. Even for common surgeries, such as a standard elective laparoscopic cholecystectomy, opioid prescriptions from physicians can vary [11].

In urology, the over-prescription and variation in prescription patterns of opioid medications is not uncommon. A recent 2019 study illustrated that following major open and minimally invasive urologic procedures, 60 percent of prescribed opioids were unused by patients based on patient surveys, indicating a discrepancy between opioid prescription and opioid necessity [12]. Other recent literature has also shown significant variation in opioid dosing among urologists; however, less is known about variation in opioid prescription refill patterns among different common urological procedure types [13,14]. We aimed to better understand prescription variability and prescription patterns following common urologic procedures.

## 2. Materials and Methods

### Population and Prescription Definition

We used data from the Truven Health MarketScan^®^ Database to identify claims from adult patients 18 years or older with either commercial or Medicare supplemental insurance that underwent common urologic procedures (Appendix A) with active healthcare insurance enrollment for 90 days following their procedure within the Austin–Round Rock, Texas metropolitan service area from 2012–2015. The Truven database is an administrative health care billing database of all claims received by payers for covered patients, including outpatient prescriptions, clinical utilization records, and healthcare expenditures [15]. We stratified the common urologic procedures into the following sub-categories: reconstructive, transurethral bladder, major oncologic—kidney, major oncologic—pelvic, stone, prosthetic, and inguinal/scrotal. While only Medicare patients with supplemental employer-sponsored plans were included, all claims from both the sponsored and Medicare plan were included. Patients with a prior diagnosis of opioid dependence (ICD 9 code 304.0) were excluded from the study. Patients discharged to skilled nursing facilities were also excluded.

We queried the Truven outpatient drug claims table for opioid prescriptions from our cohort. We defined prescription in this sense as a medication that was not only prescribed, but also filled for the patient. We excluded all unfilled prescriptions for the purposes of this study. We included prescriptions for drugs with opioid National Drug Codes (Appendix A). We analyzed these data for the total number of prescriptions, mean number of days to fill a prescription from the surgical date, and total days’ supply. The names and types of opioid medications listed were compiled from a standardized opioid classification derived from a series of clinical studies comparing the similarities and differences between existing classes of opioids [16,17]. The Truven database only maintains information on the name and number of opioid prescriptions associated with an encounter. Therefore, we calculated a minimum and maximum milligram morphine equivalent (MME) for each of the medications listed in Appendix A based on the recommended dosages from www.drugs.com and used these values as a proxy for the MME that might be associated with an encounter.

## 3. Analyses

Multivariable two-stage logistic regression and linear hurdle models were used to investigate opioid prescription outcomes, adjusting for the encounter and patient characteristics to account for the large number of patients who received no opiates. Using these quantitative models, we sought to compare opiate prescription patterns among the 7 most common types of urologic procedures. To investigate this question, we first used a logistic regression model with a binary outcome of any opioid prescriptions followed by three secondary logistic regression models that included only patients with opioid prescriptions with the following outcomes: log transformed N opiate fills, log transformed total days’ supply, and number of days from surgery to first opiate prescription claim. Age, sex, insurance plan type (PPO or other), employment status (full-time or not full-time), type of visit (inpatient or outpatient), length of stay (in days), and type of urologic procedure were included as covariates in all models.

Chi-squared and t-tests were used to assess differences in encounter, patient, and opioid prescribing characteristics among the urological procedure types. An analysis of variance (ANOVA) was used to assess the variability among opiate prescription patterns. Two-sided *p*-values less than 0.05 were considered statistically significant. All analyses were conducted using R (version 3.5.1, Foundation for Statistical Computing).

## 4. Results

We identified a total of 2312 patients who underwent urological surgery in the Austin–Round Rock Metropolitan Area between 2012 and 2015. Table 1 lists the baseline characteristics of the population. Male patients accounted for 70.6% of the total population. The mean age was 54 years. The majority of patients had surgery in an outpatient setting (90.1%) and were covered by commercial insurance (78.3%).

The most common type of urological procedure in our cohort of patients was transurethral bladder procedures (53.6%), followed by stone procedures (26.3%; Table 1). For the total population of patients both receiving and not receiving opiates, the mean length of stay (standard deviation, SD) was 0.31 days (±1.41), the mean number of days before an opiate prescription was filled was 10.26 days (±21.16), and the mean total days’ supply of opiates was 6.20 days (±23.41).

Of the total population, 549 patients (23.7%) received opiates (Table 2). Most of these patients (66.7%) were male and the most common procedure types among this population were transurethral bladder (55.7%) and stone procedures (23.9%). The mean length of stay was 0.45 days and the mean number of opiate prescriptions was 2.13 (±2.04). The mean number of days to filling an opiate prescription was 43.21 days (±21.48). Furthermore, the total days’ supply of opiates was 26.13 (±42.29).

The proportion of patients receiving opiates varied significantly by procedure type (*p* = 0.028, Table 3). From the total population in our study, patients undergoing reconstructive procedures had the highest proportion of opiates received and the highest number of mean opiate prescriptions among the seven procedure types (42% received opiates (*p* = 0.028) and the mean number of opiate prescriptions was 1.0 (*p* = 0.026); Table 3). Patients undergoing major oncologic pelvic procedures had the fewest number of associated opiate prescriptions (25.4% received opiates, *p* = 0.028, and the mean opiate prescriptions was 0.4 among all patients, *p* = 0.026). There was no difference in the mean number of days to fill opiate prescriptions and total days’ supply among urological procedures.

Our first hurdle multivariable logistic regression model with the outcome of any opiates demonstrated that no procedure types were associated with an increased odds of receiving an opioid prescription. Among those who received an opiate prescription, reconstructive patients had a shorter time to fills (mean −18.4 days, CI (−8.4–−28.5, *p* < 0.001)). In addition, among the patients that received opiates, women were found to have a greater number of opiate prescriptions (*p* = 0.02), total days’ supply (*p* = 0.00), and a shorter time to fill than men, CI (0.92–9.10, *p* = 0.02). While there was a trend towards an increased odds of receiving opiates for patients undergoing reconstructive procedures compared to other procedure types, this relationship was not significant (odds ratio (OR) 1.86, CI (0.98–3.50, *p* = 0.05)). Lastly, inpatient visits were associated with a decreased odds of receiving an opiate prescription (OR 0.58, CI (0.35–0.99, *p* = 0.04)).

## 5. Discussion

Our results suggest that there is a difference in opiate prescription patterns among different urological procedures in a major metropolitan statistical area in central Texas. While most urological procedure categories were not predictive of opioid prescribing outcomes, patients who underwent reconstructive procedures filled a larger number of opiate prescriptions when compared to other procedure categories. Thus, our results suggest that there are factors related to patients receiving reconstructive procedures that lead to elevated opioid prescription patterns when compared to the other procedure types investigated in our study. Whether this is related to the invasive nature of reconstructive surgeries or if there are other patient-level factors contributing to the prescriptive patterns is unclear (Table 4). Patients undergoing major oncologic pelvic procedures had the lowest number of opiate prescriptions, which might be explained by the fact that these patients may stay in the hospital longer (Table 3) and undergo enhanced recovery after surgery (ERAS) protocol; thus, they may have less of a need for narcotic medications upon discharge [18,19].

While this data set was not capable of distinguishing if patients were on an ERAS protocol, existing literature indicates that the optimization of peri-operative nutrition, early mobilization, and a multimodal pain management plan can lead to a reduction in the length of stay for patients and, consequently, reduced costs associated with hospitalization [18]. Perhaps incorporating elements of ERAS into outpatient or transurethral surgery would reduce narcotic dependence.

We found a difference in opiate prescribing patterns between genders, with women being more likely to receive a higher total days’ supply of opiates and refill sooner than men, which appears consistent with current literature [20]. It has been noted that women have increased pain sensitivity and a greater prevalence of painful diseases reported [21].

In another study, Shah et al. analyzed the rates and risk factors for opioid dependence or overdose after urological procedures between the years of 2007 and 2011 [13]. Independent risk factors for opioid dependence or overdose after such procedures included younger age, carriers of non-private health insurance, a longer duration of hospital stay, chronic obstructive pulmonary disease, a history of depression, or undergoing inpatient surgeries. The authors did not study reconstructive procedures specifically but did find that stone surgeries had the highest rates of 90 and 365-day opioid dependence [13]. Our results demonstrate that nearly 80% of patients undergoing procedures for kidney stones did not receive any opiates; however, our results are limited to a 90-day follow-up period. The lack of opiate prescriptions could be explained by a higher number of patients receiving pain control with various anesthetic agents intra- or post-operatively and/or non-steroidal anti-inflammatory agents post-operatively, as opposed to opiates [22].

Ziegelmann et al. sought to identify variation in opioid prescribing patterns by retrospectively analyzing these patterns in opioid-naïve patients [20]. In contrast to our study, they looked for prescriptions written within 90 days before surgery and 30 days after discharge. Consistent with our results, they discovered a significant variation in opiate prescription patterns across various types of urologic procedures. Although these results are in line with the those reported by Brummett et al., there appears to be no difference in the incidence of persistent opioid use with minor and major surgical procedures. The variation in opiate prescription patterns appears to be multifactorial and perhaps due to more patient-level factors [23]. Our study concluded that only a smaller number of patients undergoing procedures for kidney stones received an opiate prescription. This difference could be attributed to the distribution and types of procedures studied in addition to the particular patient demographics of those undergoing these specific procedures. Higher-quartile opiate prescriptions were associated with males, younger patients, those with a cancer diagnosis, and lengthened hospital stays [20]. Although our study did not analyze the size of the opiate prescriptions, our second hurdle logistic regression model results suggested that female patients and younger patients received a higher number of fills when compared to older male patients.

This study had some limitations. First, due to the content limitations of our database, we were unable to determine the specific opioid dosage information for the encounters we analyzed across all the urological procedure types. The Truven data set is an administrative billing dataset; therefore, it only includes procedures and opiates that were coded and billed to payers. It is possible some procedures were miscoded or not included accurately in our dataset, and due to the heterogeneity of our cohort, this may bias our study. In addition, although we discovered specific opioid prescription patterns for the various types of procedures, we could not determine how many of the prescribed opioids were actually consumed by the patient. Furthermore, our data set was limited to the Austin–Round Rock, TX metropolitan service area, where demographic characteristics and opioid prescribing practices might not be reflected similarly nationally. The mean number days to an opiate prescription in our study was 10.26 days, with a larger standard deviation. This could be due to providers ordering medications before a procedure date; however, we were unable to obtain any greater level of detail from our claims dataset as that described previously. Lastly, our data were retrospective in nature and were not immune to confounding errors, bias, or inaccurate bookkeeping. Despite our limitations, we were able to effectively categorize procedure types, analyze a longer post-operative follow-up period, and study multiple outcomes related to opiate prescription patterns to ultimately determine statistically significant differences between procedures in regard to opiate prescriptions.

With our results and limitations in mind, future research should ascertain the specific opioid medication and dosages of patients undergoing these urologic procedures to more precisely determine variability in opioid prescribing characteristics. We believe that by showcasing these prescriptive patterns, a call to action for standardized opiate prescription patterns is warranted [24]. Although our time frame only encompassed opioid prescribing practices between 2012–2015, it is quite possible that the patterns of prescription could have changed in more recent years due to general changing practices when it comes to prescribing narcotic medications, as well as to a different set of urologists practicing during this time. Furthermore, knowing the rates of opioid consumption by patients can suggest where opioids may or may not be more effective as a pain management modality. Lastly, considering the lack of evidence-based guidelines surrounding post-operative opioid prescriptions, future studies should aim to determine alternative pain management regimens and use validated patient-reporting measures to determine their effectiveness in resolving pain.

## 6. Conclusions

Our study of a large metropolitan area demonstrated that there was variability in opiate prescription patterns among common urological procedures. As healthcare providers, it is our responsibility to mitigate the risks associated with the over-prescription of narcotic medications, one source of which is known to be variability in prescribing patterns. The standardization and implementation of postoperative pain regimens for patients undergoing common urological procedures may help curtail this variability.

## Figures and Tables

**Table 1 jcm-11-01329-t001:** Baseline characteristics of the total population of encounters *.

Sample Size (*n*)		2312
Year (%)	2012	1074 (46.5)
	2013	503 (21.8)
	2014	444 (19.2)
	2015	291 (12.6)
Total Encounter Pay (mean (sd))		2030.31 (2697.72)
Sex (%)	Male	1632 (70.6)
	Female	680 (29.4)
Age (mean (sd))		53.89 (15.44)
Procedure Type (%)	Transurethral Bladder	1240 (53.6)
	Inguinal/Scrotal	256 (11.1)
	Major Oncologic—Kidney	83 (3.6)
	Major Oncologic—Pelvic	63 (2.7)
	Prosthetic	17 (0.7)
	Reconstructive	45 (1.9)
	Stone	608 (26.3)
Length of Stay (mean (sd))		0.31 (1.41)
Number of Opiate Prescriptions (mean (sd))		0.50 (1.34)
Mean Days to Opiate Prescriptions (mean (sd))		10.26 (21.16)
Total Days’ Supply of Opiates (mean (sd))		6.20 (23.41)
Any Opiates Received (%)	No	1763 (76.3)
	Yes	549 (23.7)
Total Min MME (mean (sd))		2.62 (8.91)
Total Max MME (mean (sd))		15.23 (49.93)
Visit Type (%)	Inpatient	230 (9.9)
	Outpatient	2082 (90.1)
Payer (%)	Commercial	1811 (78.3)
	Medicare	501 (21.7)
Plan Type (%)	Non PPO	516 (23.1)
	PPO	1718 (76.9)
Employment Status (%)	Full-Time	989 (42.8)
	Not Full-Time	1323 (57.2)

* Excludes opiate use disorder patients and patients with repeat procedures in the same category within 90 days.

**Table 2 jcm-11-01329-t002:** Baseline characteristics of the population of encounters receiving opiates.

Sample Size (*n*)		549
Year (%)	2012	282 (51.4)
	2013	106 (19.3)
	2014	110 (20.0)
	2015	51 (9.3)
Total Encounter Pay (mean (sd))		2006.91 (2873.67)
Sex (%)	Male	366 (66.7)
	Female	183 (33.3)
Age (mean (sd))		53.49 (15.14)
Procedure Type (%)	Transurethral Bladder	306 (55.7)
	Inguinal/Scrotal	51 (9.3)
	Major Oncologic—Kidney	20 (3.6)
	Major Oncologic—Pelvic	16 (2.9)
	Prosthetic	6 (1.1)
	Reconstructive	19 (3.5)
	Stone	131 (23.9)
Length of Stay (mean (sd))		0.45 (1.56)
Number of Opiate Prescriptions (mean (sd))		2.13 (2.04)
Mean Days to Opiate Prescription Fill (mean (sd))		43.21 (21.48)
Total Days’ Supply Opiates (mean (sd))		26.13 (42.29)
Any Opiates Received (%)	No	0 (0.0)
	Yes	549 (100.0)
Total Min MME (mean (sd))		11.68 (15.76)
Total Max MME (mean (sd))		67.86 (86.88)
Visit Type (%)	Inpatient	72 (13.1)
	Outpatient	477 (86.9)
Payer (%)	Commercial	433 (78.9)
	Medicare	116 (21.1)
Plan Type (%)	Non PPO	122 (22.9)
	PPO	410 (77.1)
Employment Status (%)	Full-Time	214 (39.0)
	Not Full-Time	335 (61.0)

**Table 3 jcm-11-01329-t003:** Opiate prescription patterns among reconstructive urologic procedures compared to other urologic procedures.

Category of Procedure		Reconstructive	Transurethral Bladder	Inguinal/Scrotal	Major Oncologic—Kidney	Major Oncologic—Pelvic	Prosthetic	Stone	*p*-Value
Number of Encounters		45	1240	256	83	63	17	608	
Number of Opiate Prescriptions (mean (sd))		1 (1.8)	0.5 (1.4)	0.4 (1.3)	0.7 (2.4)	0.4 (0.7)	0.8 (1.5)	0.4 (1.1)	0.026
Mean Days to Fill Opiate Prescription (mean (sd))		10.8 (17.2)	11.2 (22)	7.5 (17.8)	9.7 (20.3)	7 (15.4)	14 (24.7)	9.8 (21.5)	0.164
Total Days’ Supply of Opiates (mean (sd))		9.7 (30.7)	7.1 (25.3)	2.6 (8.9)	7.4 (31.3)	4.2 (13.4)	10.6 (32.4)	5.6 (22.0)	0.105
Any Opiates Received	No	26 (57.8)	934 (75.3)	205 (80.1)	63 (75.9)	47 (74.6)	11 (64.7)	477 (78.5)	0.028
	Yes	19 (42.2)	306 (24.7)	51 (19.9)	20 (24.1)	16 (25.4)	6 (35.3)	131 (21.5)	

**Table 4 jcm-11-01329-t004:** Demographic information among patients who undergo reconstructive urologic procedures compared to other urologic procedures.

Category of Procedure	Reconstructive	Transurethral Bladder	Inguinal/Scrotal	Prosthetic	Stone	Major Oncologic—Kidney	Major Oncologic—Pelvic
Sex (Male % (Female %))	31.1 (68.9) *	73.1 (26.9)	99.2 (0.8)	94.1 (5.9)	53.5 (46.5)	72.3 (27.7)	90.5 (9.5)
Age in Years (mean (sd))	46.6 (11.4) *	58.3 (14.8)	44.2 (15.9)	63.4 (11.2)	48.5 (13.9)	55.8 (12.1)	58.5 (9.8)
Payer Type (Commercial (Medicare))	93.3 (6.7) *	71 (29)	91.4 (8.6)	52.9 (47.1)	88 (12)	75.9 (24.1)	76.2 (23.8)
Plan Type (Non PPO (PPO))	6.8 (93.2) *	26.7 (73.3)	17.3 (82.7)	29.4 (70.6)	19.1 (80.9)	24.4 (75.6)	23.7 (76.3)
Encounter Pay (mean $ (sd $))	1755 (1088) *	1045 (1790)	1715 (2250)	4329 (4525)	3912 (3357)	2825 (2497)	2078 (1931)

* *p*-value ≤ 0.001.

## Data Availability

Restrictions apply to the availability of these data. Data was obtained via the MarketScan database from Victoria Valencia and are available with permission of Victoria Valencia.

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
