# Peer review of "Opiate Prescriptions Vary among Common Urologic Procedures: A Claims Dataset Analysis"

_jcm, 2022, doi:10.3390/jcm11051329_

Round 1
Reviewer 1 Report
No major comments
Author Response
Point 1: English language and style are fine/minor spell check required. No major comments
Response 1: We thank you for the feedback. All grammatical errors have been correct to the best of our ability.
Reviewer 2 Report
Congrats for your work. Nevertheless there are some issues that you have correct.
The author response to the editor says: "Furthermore, MarketScan data is only complete through 2018 currently so we do not feel that the longitudinal addition of newer and incomplete data from MarketScan would add much to this study"
I feel that the longitudinal adition of data until 2019 would add some information to the study. I recommend to extend the analysis period.
In discussion section first paragraph, authos say: 264 "Our study of a large metropolitan area demonstrated that there is a variability in opiate prescription patterns among common transurethral urological procedures"
PLease, consider to change to "Our study of a large metropolitan area demonstrated that there WAS a variability in opiate...."
Best regards
Author Response
Point 1: The author response to the editor says: "Furthermore, MarketScan data is only complete through 2018 currently so we do not feel that the longitudinal addition of newer and incomplete data from MarketScan would add much to this study" I feel that the longitudinal adition of data until 2019 would add some information to the study. I recommend to extend the analysis period.
Response 1: We appreciate the reviewer’s critiques. As mentioned previously, we are unable to obtain quality data up until this point and this has been discussed in our limitations already.
Point 2: In discussion section first paragraph, authos say: 264 "Our study of a large metropolitan area demonstrated that there is a variability in opiate prescription patterns among common transurethral urological procedures" PLease, consider to change to "Our study of a large metropolitan area demonstrated that there WAS a variability in opiate...."
Response 2: Thank you for the feedback. The reviewer’s suggestion has been incorporated in our paper accordingly.
Round 2
Reviewer 2 Report
Thank you for your work.
This manuscript is a resubmission of an earlier submission. The following is a list of the peer review reports and author responses from that submission.
Round 1
Reviewer 1 Report
Materials and methods: probably you have counted some patients twice, for example 1st stage urethroplasty or 1st stage hypospadia and buccal mucosa harvest (Table 1). Please, provide a suitable explanation.
The collected data are related to 2012-2015. Do you think there is a modification of type of prescription in 2018-2021?
Results page 7: "The mean number of days to filling an opiate prescription was 43.21 days (+/- 21.48)". How do you know the prescription is related to the surgical procedure and not to the previous condition of the patient?
Discussion section: I think these paragraphs should go to results section:
"The proportion of patients receiving opiates varied significantly by procedure type (p=0.028, Table 3)"
"While the majority of urological procedure categories were not pre- 170
dictive of opioid prescribing outcomes, patients who underwent reconstructive procedures filled a larger number of opiate prescriptions when compared to other procedure categories"
Conclusions in abstract and conclusion's section are different
Reviewer 2 Report
The aim of the study was to describe total day supply and fills of common opiates following urologic procedures. The study is cahotic, primary and secondary outcomes not fully reported. Prescription definitions were not reported, making difficult to read the manuscript. Moreover, there are too many surgical procedures, for different pathologies which are difficult to compare. There are several critical aspects that author need to review in order to improve overall quality of manuscript.
Round 2
Reviewer 1 Report
Thank you for your answer.
The manuscript is now ready in form. Nevertheless is based on an old database (2012-2015) so the conclusions probably are not applicable to our current clinical practice.
Reviewer 2 Report
No major comments.